# Biomimetic Study of a Honeycomb Energy Absorption Structure Based on Straw Micro-Porous Structure

**DOI:** 10.3390/biomimetics9010060

**Published:** 2024-01-21

**Authors:** Shucai Xu, Nuo Chen, Haoyi Qin, Meng Zou, Jiafeng Song

**Affiliations:** 1State Key Laboratory of Intelligent Green Vehicle and Mobility, Tsinghua University, Beijing 100084, China; xushc@tsinghua.edu.cn; 2Suzhou Automobile Research Institute (Xiangcheng), Tsinghua University, Suzhou 215133, China; 13832240299@163.com (N.C.); qin15517537994@163.com (H.Q.); 3College of Mechanical and Electrical Engineering, Hebei Agricultural University, Baoding 071001, China; 4Key Laboratory for Bionics Engineering of Education Ministry, Jilin University, Changchun 130022, China; zoumeng@jlu.edu.cn; 5Key Laboratory of Transportation Industry for Transport Vehicle Detection, Diagnosis and Maintenance Technology, Jinan 250357, China

**Keywords:** straw, porous structure, honeycomb structure, energy-absorbing structure, bionic design

## Abstract

In this paper, sorghum and reed, which possess light stem structures in nature, were selected as biomimetic prototypes. Based on their mechanical stability characteristics—the porous structure at the node feature and the porous feature in the outer skin— biomimetic optimization design, simulation, and experimental research on both the traditional hexagonal structure and a hexagonal honeycomb structure were carried out. According to the two types of straw microcell and chamber structure characteristics, as well as the cellular energy absorption structure for the bionic optimization design, 22 honeycomb structures in 6 categories were considered, including a corrugated cell wall bionic design, a modular cell design, a reinforcement plate structure, and a self-similar structure, as well as a porous cell wall structure and gradient structures of variable wall thickness. Among them, HTPC-3 (a combined honeycomb structure), HSHT (a self-similar honeycomb structure), and HBCT-257 (a radial gradient variable wall thickness honeycomb structure) had the best performance: their energy absorption was 41.06%, 17.84%, and 83.59% higher than that of HHT (the traditional hexagonal honeycomb decoupling unit), respectively. Compared with HHT (a traditional hexagon honeycomb decoupling unit), the specific energy absorption was increased by 39.98%, 17.24%, and 26.61%, respectively. Verification test analysis revealed that the combined honeycomb structure performed the best and that its specific energy absorption was 22.82% higher than that of the traditional hexagonal structure.

## 1. Introduction

Energy absorption devices are widely used in automobiles, ships, airplanes, railway trains, and other vehicles and are the main components to dissipate impact kinetic energy in the event of a collision or other emergencies [1,2]. In the event of a traffic accident resulting in a collision, an energy absorption device can protect machinery from serious damage when subjected to an impact load, minimizing the injury to humans. These energy-absorbing devices disperse kinetic energy in a variety of ways, including friction, fracture, pressure, plastic bending, and cyclic plastic deformation [3]. The thin-walled metal tube is the most widely used energy-absorbing element at present. Research has shown that, after scientific design, the single thin-walled tube structure has a controllable failure mode, a relatively stable compression load, and is an excellent buffer energy-absorbing element [4]. However, with the increase in lightweight materials and the improvement of safety requirements, the optimization of the energy absorption performance of metal thin-walled tubes faces theoretical, methodological, and technical challenges. The design of a lightweight, efficient, and successful crash-absorbing thin-wall structure has practical significance for protecting human life and personal property, as well as implications for saving energy and protecting the environment. The honeycomb structure has the best lightweight performance among the energy-absorbing structures [5,6,7].

Inspired by a variety of biological structures in nature, bionic structures have significantly improved energy absorption capacity compared with traditional structures. Therefore, in recent years, the use of biomimetic methods to design new lightweight structures with excellent energy absorption capacity has increased. In nature, various plants and animals possess many low-density, high-strength, high-energy-absorbing structures, which have provided inspiration for the human design of energy-absorbing structures with better performance [2,8,9,10,11,12,13,14,15]. Recently, much research has focused on the honeycomb structure. Lin et al. [16] designed a novel honeycomb structure with twisted features manufactured via laser powder bed fusion (LPBF), which was inspired by the honeycombs produced by bees. Their results revealed that the structure with a 0.75 mm wall thickness and three unit cells along each side demonstrated the highest specific energy absorption ability. Zhang [17] proposed a bio-inspired re-entrant arc-shaped honeycomb (RAH) model. Due to the introduction of re-entrant arc-shaped structures, the dynamic response curves of bio-inspired RAHs have better crushing load uniformity than conventional re-entrant honeycombs. Lightweight auxetic reentrant honeycombs (ARHs) with a negative Poisson’s ratio (NPR) are very promising for crashworthiness applications due to their high specific strength and energy absorption (EA). To enhance the potential of ARHs, a bio-inspired self-similar “concentric auxetic reentrant honeycomb (CARH)” was proposed by Jiang [18]. There is a theory that bio-inspired CARHs show higher plateau stresses and EA than traditional ARHs. Inspired by the microstructure of pomelo peel, Zhang [19] proposed a novel hierarchical honeycomb and investigated its crushing resistance along with its energy absorption capabilities. Simulations revealed that the deformation modes of bio-inspired honeycombs are governed by geometric parameter-equivalent thickness. Bio-inspired honeycombs present a novel perspective on the mechanical properties of natural cellular materials. There are also other bionic honeycomb structures that exhibit excellent mechanical properties, such as bamboo [20,21], beetle elytron [22,23], horseshoe [24], turtle shell [25], ladybeetle [26], horn [27,28], and so on.

In nature, there are many stems with stable structural and mechanical properties, such as bamboo, reed, sorghum, and cattail stems, whose slenderness ratio can reach 1/100 ~ 1/270. Such a slender structure ensures that the stem will not be damaged by loads in nature, which is difficult for conventional structures to achieve. Most of these slender stem plants have nodal features, which can enhance stem bending strength, radial extrusion, and shear resistance. Slender stem plants with nodes can be roughly divided into those with a hollow structure and those with a solid structure. Hollow stems have a hollow structure inside. A solid stem differs from a hollow stem in that it has a distinct inner core and a continuous tubular outer sheath distributed outwards from a light, foam-shaped center. 

Typical straw structures possess some of the best mechanical properties in nature. Research shows that straw has higher shear and compression resistance than other stem plants, especially given that its lodging strength depends on its bending strength [29,30,31]. Such straw structures have many similarities in structure, function, and load form with thin-walled structures, which can both provide inspiration and serve as a reference for lightweight and crashworthy designs using thin-walled structures. For example, sorghum straw has a compound filling structure. The dermal tissue on the wall of the stem and tube is highly dense, mainly composed of small and dense fibrous bundles. The inner medullary core is a porous foam structure, and its function is similar to that of the foam core [32]. The characteristic periodic nodes along the growth direction of the stem also enhance the stem’s resistance to deformation [33,34]. A sorghum stalk is round in cross-section and oval when cut, forming a reinforcing ring structure similar to that of a double-ring groove. Within the stem, there are large vascular bundles (used to transport water and nutrients) with a foamy matrix tissue between them. The interaction between vascular bundles and the foam matrix not only provides stronger support for the stem but also effectively reduces the mass of the structure. Another common slender stalk is the reed. Phragmites australis is a monocotyledonous plant belonging to the *Gramineae* family [35]. A reed rod is usually a slender member, mainly used to bear external loads and dead weight. Its structure presents a gradual structure from inside to outside, and its elongation and gradient characteristics play an important role in the stability and bearing capacity of this structure [36,37]. Therefore, this paper examines two kinds of straw with nodes in nature, sorghum (solid) and reed (hollow), as biomimetic prototypes to carry out biomimetic lightweight research on honeycomb energy-absorbing structures, as shown in Figure 1.

## 2. Bionic Design of Honeycomb Structures

For these two kinds of straw stalk structures, the mechanical properties of light weight and high strength are directly related to the macro- and micro-structural characteristics. As straw is a high-fiber structure, the integrity of the section structure cannot be guaranteed using ordinary cutting tools. Therefore, after the sample was frozen in liquid nitrogen for 5 h, the straw was cut horizontally with a sharp blade. A Zeiss scanning electron microscope (SEM, Model EVO-18, Germany) was employed. The main parameters of this SEM were as follows: the experimental magnification range is 13–50,000 times; the minimum resolution is 3.0 nm; and the instrument utilizes a tungsten filament. The transverse and longitudinal typical structural characteristics of each part of the stalk of the two kinds of straw obtained are shown in Figure 2a–e.

### 2.1. Bionic Design of the Cell Edge of a Honeycomb

According to the electron microscope observation of reed straw, the porous structure of reed in axial and longitudinal sections is similar to the honeycomb structure. The crystal cells at the node feature are denser, as shown in Figure 2a. The boundary morphology of the crystal cells is similar to the sideline structure of the sinusoidal curve. As can be seen from the measurement, the curve structure can be expressed as follows:(1)$$y=sin\left(3.24\times x+\pi /2\right)+21$$
where *x* is the transverse width of the curve, and *y* is the longitudinal height of the curve.

Based on the principles of engineering bionics, bionic sinusoids were applied to the edge lines of hexagonal thin-walled structures, hexagonal honeycomb structures, quadrilateral honeycomb structures, and quadrilateral thin-walled structures. At the same time, for comparative analysis, the corresponding linear thin-wall and honeycomb structures were established, as shown in Figure 3a–d. The wall thickness of all the structures was 0.5 mm, and they are named: Hexagon Tube with Bionic Corrugated (HTBC), Hexagon Honeycomb Tube with Bionic Corrugated-7 (HHTBC-7), Hexagon Honeycomb Tube with Bionic Corrugated (HHTBC), Square Tube with Bionic Corrugated (STBC) (STBC), Hexagon Tube (HT), Hexagon Honeycomb Tube-7 (HHT-7), Hexagon Honeycomb Tube (HHT), and Square Tube (ST).

### 2.2. Bionic Design of Honeycomb Cell Compound Structure

In the stem structure of the reed, there are nodal features similar to the bamboo nodal structure. The nodule structure on *Phragmites communis* also enhanced its stem structure, which was inevitably related to its microstructure. Combined with the observation of the microstructure, it is found that there is a special combined structure at the node, as shown in Figure 2b. In the porous structure of the nodes, there are compound structures with special links of pentagons and circles, which are widely distributed in the parts of the nodes. Through measurement statistics, it is found that the ratio of the outer circle to the circle of the pentagon is R1:R2, which is nearly 5:1. Based on this, this paper proposed three kinds of bionic composite cell honeycomb structures, which were named Honeycomb Tube with Pentagon and Circular-1, -2, and -3 (HTPC-1, -2, and -3), as shown in Figure 3e–g.

### 2.3. Bionic Design of Honeycomb Cell Stiffener Ribs

When the reed stem structure is subjected to transverse loads, its stem wall plays the main bearing role. In our observation of the microstructure, we found that there were regular octagonal structures on the tube wall, and the octagonal nodes were connected by the cell wall structure. In this paper, this structure is simplified to the honeycomb structure, as shown in Figure 2c. The octagonal cell structure serves as the fundamental unit, with the octagonal cell connected to the reinforcement ribs. It is denoted as Octagonal Honeycomb Tube (OHT) and Octagonal Honeycomb Tube with Bionic Ribs (OHTBR), depicted in Figure 3h,i.

### 2.4. Self-Similar Bionic Design of a Honeycomb Cell

For sorghum straw, its inner stem is porous and composed of a vascular bundle and a foam matrix. The vascular bundle distribution was denser at the feature of the sorghum node, and there was a specific morphology. The internal vascular bundle showed hierarchical self-similarity through SEM observation and analysis, as shown in Figure 2d. The cell structure around the vascular bundle is usually centered on a larger cell structure, radiating and showing some similarity. Therefore, to simplify this analysis, we designed the following two levels of self-similar thin-wall and honeycomb structures, as shown in Figure 3j,k, and named them Hierarchical Self-similar Tube (HST) and Hierarchical Self-similar Honeycomb Tube (HSHT), respectively.

### 2.5. Cellular Bionic Porous Design

In the longitudinal fiber structure of sorghum straw, such a structure also exists, with a large number of small holes distributed on the wall of the tube. The existence of such a hole structure is conducive to the full transport of water to each part of the stem. On the other hand, it may also be an optimal design for weight reduction, as shown in Figure 2e. Based on this, the following bionic porous structures are proposed and named: Circular Tube (CT), Circular Tube with Bionic Holes (CTBH), Circular Honeycomb Tube (CHT), and Circular Honeycomb Tube with Bionic Holes (CHTBH), as shown in Figure 3l–o.

### 2.6. Cellular Gradient Bionic Design

In addition, the porous structure of the above two kinds of stems has a common point, which is the gradient change trend. There is a gradient change in both the size of the cell structure and the thickness of the cell. Based on this, this paper proposes a bionic radial honeycomb structure with variable wall thickness, as shown in Figure 3p–r. They are divided into two types: one with the inner wall thickness gradually decreasing from the inside out and from the outside, and the other being the honeycomb structure with equal wall thickness, named Honeycomb Bionic Variable Thickness (HBVT-257), Honeycomb-Uniform Thickness (HUT/HBVT-555), and Honeycomb Bionic Variable Thickness (HBVT-752).

The mass of the hammer used in this experiment was 100 kg for each specimen, and the impact velocity was 6.85 m/s. In order to facilitate comparative analyses and transverse comparisons of the energy absorption characteristics of each honeycomb structure, the total mass of each honeycomb structure was controlled to be the same in this paper. Table 1 shows the energy absorption characteristics of each honeycomb structure parameter and axial impact load.

## 3. FEA Analysis

### 3.1. The FE Model

The nonlinear finite element software tools Hypermesh and LS-DYNA were used for simulation analysis. In the Optistruct module of Hypermesh software, tetrahedral meshing was performed on the honeycomb energy absorption structure. After adjusting the element sizes, splitting, and other operations, a mesh with approximately 120,000 nodes and 100,000 non-distorted C3D4 elements was obtained. Figure 4 shows the finite element model of the axial impact honeycomb structure. Both the ground and impact surfaces are rigid structures, so they are regarded as rigid bodies. The “face-to-face” contact algorithm with a friction coefficient of 0.3 was used to simulate the contact between the rigid wall and the honeycomb specimen. The “automatic single surface” contact method was adopted to regulate the honeycomb structure itself to avoid the mutual penetration of bending in the process of bending failure. For comparative analysis, the compression distance of all the honeycomb structures was set to 80% of the sample height. The material used in this paper is the AA6061-T6 aluminum alloy, and its mechanical properties are calibrated using the standard tensile test: density 2700 kg/m^3^, Poisson’s ratio 0.3, and Young’s modulus 70 GPa, as shown in Figure 5. The constitutive model of the thin-walled tube was simulated using MAT_24 in the LS-DYNA software. Since aluminum alloy is a strain-rate-insensitive material, the strain–rate effect was not considered [38,39].

### 3.2. The Energy Absorption Index

Energy absorption (*EA*) [40,41,42], obtained by integrating the load–displacement curve during the loading process mathematically, is:(2)$$EA=\int_{0}^{x}F(x)dx$$

The higher the energy absorption *(EA*), the better the crashworthiness. To account for the effect of mass, specific energy absorption (*SEA*) [43,44,45] is defined as:(3)$$SEA=\frac{EA}{m}=\frac{\int_{0}^{x}F(x)dx}{m}$$

For the energy-absorbing structure, the higher its *SEA*, the better its capability of energy absorption.

The crushing force efficiency (*CFE*) [46,47] is another criterion in relation to structural deformation stability, which can be given as:(4)$$CFE=\frac{F_{mean}}{F_{\max}}\times 100\%$$
(5)$$F_{mean}=\frac{EA}{s}$$
where *F_mean_* is the mean loading force, and *F_max_* is the maximum loading force.

### 3.3. Results and Analysis

Figure 6 shows the load–displacement curves of the bionic honeycomb structure, and Figure 7 shows the deformation and stress cloud of the honeycomb structure under axial impact load.

As can be seen from Figure 6a, the HT and ST can easily undergo large buckling and deformation due to their monocellular structure, so their load curves fluctuate greatly. The overall performance is bottom-up folding deformation, and the folding radius of buckling deformation is large (as shown in Figure 7). However, the load curves of the HHT and HHT-7 conventional hexagonal honeycomb polycellular structures are relatively stable, and their deformation is also a bottom-up progressive folding deformation. However, due to the characteristics of polycellular structures, the folding radius is small and the stress distribution is uniform. For the corrugated structure with bionic optimization design, its deformation and buckling are unstable under axial impact, especially for the monocellular bionic bellows structure, which is prone to instability when large deformation occurs, resulting in failure deformation in the middle part (HTBC, HHTBC-7, HHTBC, and STBC in Figure 7). Therefore, its load-bearing and energy absorption characteristics are poor. The deformation and load curve of the bionic polycellular bellow tube wall structure are different from those of the single-cell structure, but the bionic design scheme is not a beneficial design method.

Figure 6b shows the comparison of load curves between the combined honeycomb structures HTPC-1, -2, and -3 and the traditional hexagonal honeycomb structure HHT. It can be seen from this figure that the load curves of HTPC-1 and HTPC-3 are slightly higher than those of the traditional hexagonal honeycomb structure HHT, while the load curve of HTPC-2 fluctuates greatly. Combined with the deformation and stress cloud in Figure 7, it can be seen that HTPC-2 has a large deformation with central shrinkage in the middle and late deformation, resulting in the overall collapse of the honeycomb structure, which reduces its bearing capacity and energy absorption effect. In contrast, HTPC-1,3 has undergone progressive folding deformation centering on a single crystal cell. The difference is that the late deformation of HTPC-1 and HTPC-3 is stable from top to bottom, while the deformation of HTPC-1 also occurs at the bottom, resulting in the disorder of the deformation order. However, the overall loading and energy absorption effects of the two are better than that of the traditional hexagonal honeycomb structure HHT.

Figure 6c shows the compression load–displacement curves of the OHT and OHTBR honeycomb structures. It can be seen that the load peaks of both are slightly lower than that of the traditional honeycomb structure HHT, and the fluctuation amplitude is also slightly less than that of the HHT, especially for the OHT. Compared with the OHT and OHTBR, it can be seen that the load curve of the bionic stiffened rib has been improved to a certain extent, and the stress distribution in the deformation process is more uniform. Notably, the stress distribution in the third stress cloud diagram of the OHTBR (as shown in Figure 7) is significantly more uniform than that of the OHT, which means that the stress distribution and deformation of the OHT are more stable.

The self-similar structure has been a research hotspot in recent years. The comparison structures proposed in this paper are the hexagonal self-similar structure HST and the hexagonal self-similar honeycomb tube HSHT. Comparing the HSHT and HHT with the same configuration, the self-similar structure has more wall structures, so its bearing capacity is obviously higher. At the same time, when more tube walls are subjected to impact loads, there will be an interaction between tube walls. Therefore, the compression folding radius is relatively small, so the load fluctuation is also small; that is, the load fluctuation is smaller (Figure 6d). Because of this, the multilayer structure of the HST makes its deformation not top-down but simultaneous deformation of the middle and lower parts, which affects its deformation stability. At the same time, the multi-layer structure makes it larger in mass. For the HSHT, the honeycomb structure itself has certain deformation stability, and its deformation stability is increased during the self-similar design of crystal cells (as shown in the HSHT in Figure 7).

Figure 6e shows the load curves of the CTBH and CHTBH and their corresponding thin-walled tubes under impact. It can be seen from this figure that the bearing capacity of the structure decreases significantly after the bionic hole structure is introduced into the tube wall. At the same time, the load fluctuation is smaller than that of the corresponding intact wall structure. Through calculation, it is found that although the bearing capacity decreases, the whole mass is greatly reduced due to the bionic hole, so the specific energy absorption of the whole is improved. Therefore, this design scheme has a certain application value in scenarios requiring weight reduction and peak load reduction.

Figure 6f is the comparison diagram of the compressive load–displacement curve of the honeycomb structure with radial gradient variable wall thickness. It can be seen from this figure that the larger the outer wall thickness, the stronger its carrying capacity will be, but at the same time, the peak value of its load and the overall mass will also greatly increase. In terms of stress distribution and deformation stability, the deformation of the gradient structure is relatively more stable and the stress distribution is more uniform, especially for HBVT-257, whose wall thickness gradually decreases from inside to outside, as shown in Figure 7.

Through a comprehensive comparison of the above structures and their bionic design samples, the evaluation indexes of crashworthiness and energy absorption in Figure 8 can be obtained through calculation. The red dotted line in this figure shows the traditional hexagonal honeycomb structure as a horizontal comparison. The analysis shows that the performance of the biomimetic design HTPC-1, HTPC-3, OHT, CHT, HSHT, and HBVT257 is better than that of the traditional hexagonal honeycomb structure. Among them, HTPC-3, HSHT, and HHT-257 have the best performance, and their energy absorption is increased by 41.06%, 17.84%, and 83.59% compared with the HHT, and 39.98%, 17.24%, and 26.61% compared with the HHT, respectively. In consideration of peak load and crushing force efficiency, we selected HTPC-3, HSHT, OHT, OHTBR, and HHT-257 as the objects of the next part of the experiment to study the optimal solutions of these bionic designs from an experimental perspective.

## 4. Experimental Study

### 4.1. Processing and Manufacturing

In order to verify the effectiveness of the bionic design, the bionic honeycomb structure with excellent performance in the simulation analysis was verified. Due to its complex structure, the 3D printer model used in this paper was the EOSINT M280 (a metal 3D printer) to print and process part of the honeycomb structure. This printer uses direct metal powder laser sintering technology to build parts layer by layer by melting fine metal powder with the laser beam. It can support the creation of extremely complex geometric components, such as free-form surfaces, deep grooves, and 3D cooling channels, and can carry out CAD interface, STL, and other format conversions. The samples in this paper were saved as an STL file in CATIA and imported into a 3D printing system. The molding size of the printer is 250 × 250 × 325 mm, the precision is 20–80 μm, and the consumable material is metal powder. This test used stainless steel metal powder; its basic parameters are: density = 7.8 g/cm^3^; Young’s modulus = 180 GPa; and yield strength = 550 MPa. The patterns of these 3D samples are shown in Figure 9.

To better observe the deformation modes of the honeycomb structures with different biomimetic cells, quasi-static axial compression tests were conducted using the ETM-300 (with a maximum load capacity of 300 kN) at the Key Laboratory of Engineering Bionics, Ministry of Education, Jilin University. All the designed samples were tested at standard room temperature (25 °C). The test loading speed was set at 3 mm/min, and the final loading displacement was 2/3 of the sample height. The load and displacement data were obtained from the data acquisition system. During this experiment, the deformation modes during compression were recorded using a camera.

### 4.2. Results and Analysis

As the wall thickness in the simulation analysis was analyzed according to the standard honeycomb structure (wall thickness: 0.02–0.08 mm), the actual 3D printing accuracy was determined to be at least 0.8 mm. At the same time, the phenomenon of material fracture is not considered in the simulation analysis, but in the real experiment, due to the thick wall and small size, the phenomenon of fracture occurred. Therefore, the verification test only makes a transverse comparison and does not compare and analyze the simulation results. Figure 10 shows the compression test process for some samples. Due to the thickness and material problems of the processed materials, most of them were crushed and fractured without significant buckling or folding.

According to the actual deformation interval, the energy absorption of the first 30% of the deformation area was calculated, and its axial compression energy absorption characteristics were obtained, as shown in Table 2. According to this table, the energy absorption capacity of the combined honeycomb structure HTPC-3 is the most excellent, which is 22.82% higher than that of the common hexagonal honeycomb structure HBVT-555.

## 5. Conclusions

In this paper, based on the characteristics of two types of straw microcells and chamber structures, a cellular energy absorption structure is proposed for bionic optimization design. A total of 22 honeycomb structures are classified into 6 categories, including a bionic design with corrugated cell walls, a modular cell design, a reinforcement plate structure, a self-similar structure, a porous structure of cell walls, and a gradient structure with variable wall thickness. Among them, HTPC-3 (combined honeycomb structure), HSHT (self-similar honeycomb structure), and HBCT-257 (radial gradient variable wall thickness honeycomb structure) have the best performance, and their energy absorption is 41.06%, 17.84%, and 83.59% higher than that of the HHT (traditional hexagonal honeycomb decoupling unit), respectively. Compared with the HHT (traditional hexagon honeycomb decoupling unit), the specific energy absorption is increased by 39.98%, 17.24%, and 26.61%, respectively. Through verification test analyses, the HTPC-3 structure (combined honeycomb structure) has the best design performance, and its specific energy absorption is 22.82% higher than that of the traditional honeycomb structure. The conclusions of this study can provide a new idea and reference for the optimization design of the honeycomb structure.

## Figures and Tables

**Figure 1 biomimetics-09-00060-f001:**
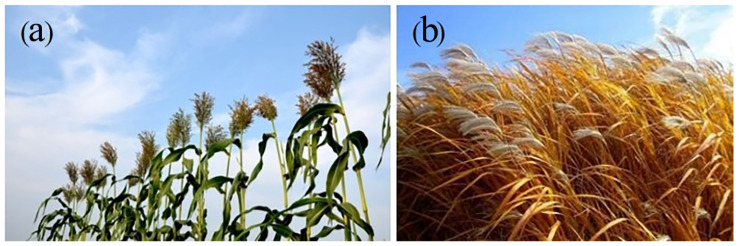
Bionic prototype: (**a**) sorghum straw; and (**b**) reed straw.

**Figure 2 biomimetics-09-00060-f002:**
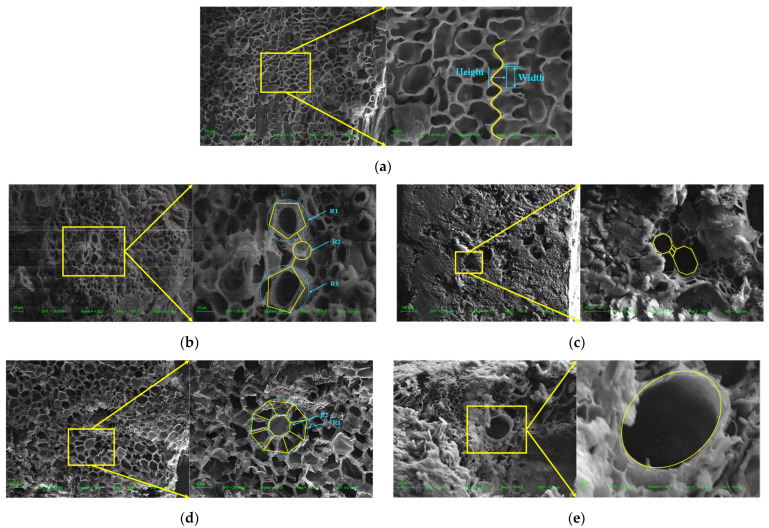
Characteristics of the microcellular structure of straw (SEM view): (**a**). crystal cell boundary line structure of longitudinal cross-section at reed node; (**b**). microstructure of vascular bundle at reed node; (**c**). microstructure of the outer sheath wall of *Phragmites australis*; (**d**). microstructure of large vascular bundle structure of sorghum; and (**e**). microstructure of small vascular bundle of sorghum straw (R1 is the inner circle radius of the large vascular bundle, and R2 is the outer circle radius of the small vascular bundle).

**Figure 3 biomimetics-09-00060-f003:**
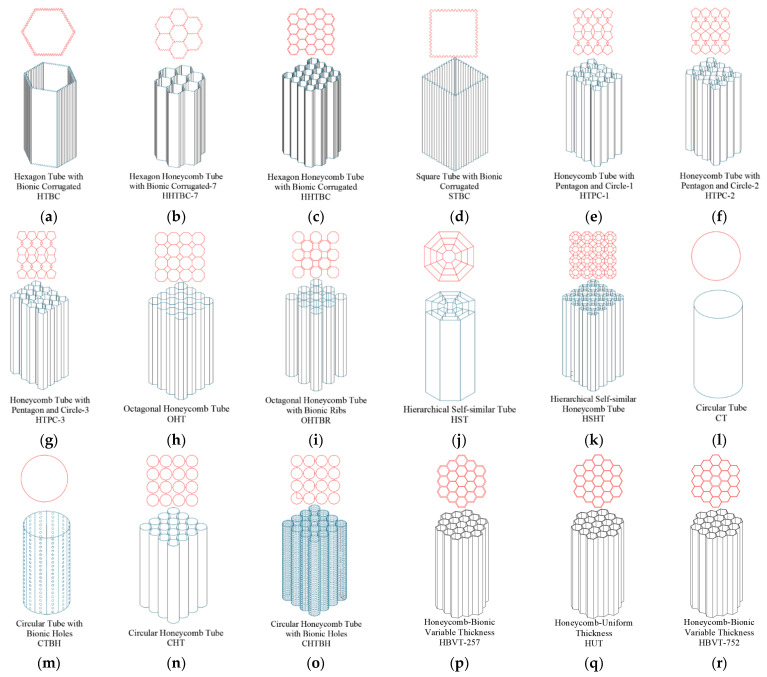
Bionic design of honeycomb structures: (**a**). HTBC; (**b**). HHTBC-7; (**c**). HHTBC; (**d**). STBC; (**e**). HTPC-1; (**f**). HTPC-2; (**g**). HTPC-3; (**h**). OHT; (**i**). OHTBR; (**j**). HST; (**k**). HSHT; (**l**). CT; (**m**). CTBH; (**n**). CHT; (**o**). CHTBH; (**p**). HBVT-257; (**q**). HUT/HBVT-555; and (**r**) HBVT-752.

**Figure 4 biomimetics-09-00060-f004:**
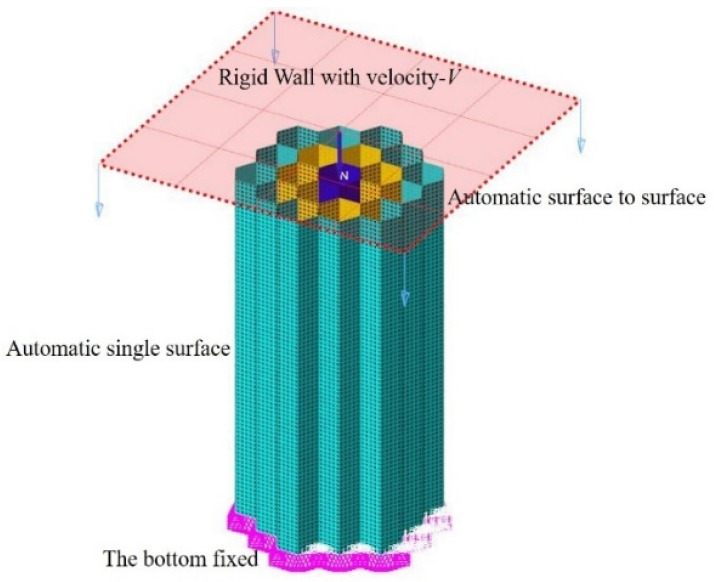
Finite element simulation analysis model (example: radial-gradient variable-wall thickness honeycomb structure). This figure uses HBVT as an example to demonstrate its finite element simulation environment, where different grid colors from the inside out represent different wall thicknesses.

**Figure 5 biomimetics-09-00060-f005:**
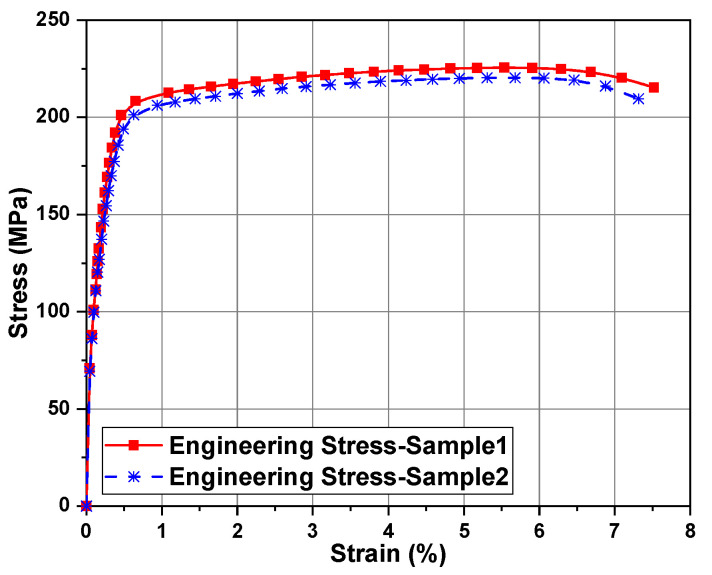
Stress–strain curve of AA6061-T6.

**Figure 6 biomimetics-09-00060-f006:**
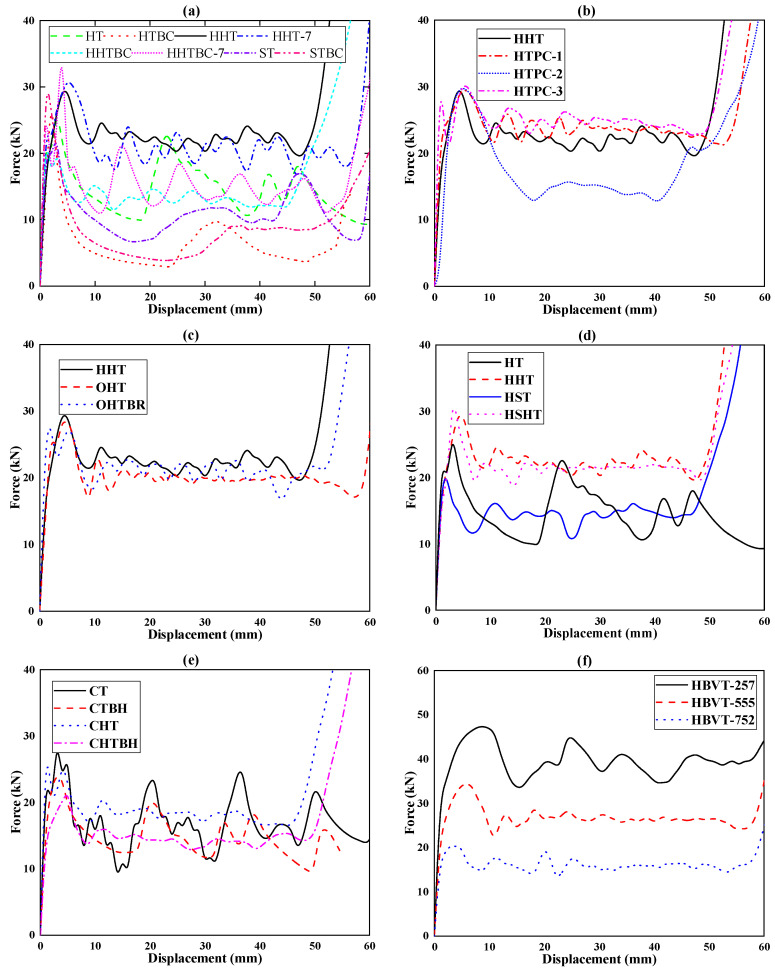
Force–displacement curves of each energy-absorbing structure under axial impact. (**a**). Load–displacement curves of the HT, HTBC, HHT-7, HHTBC-7, HHT, HHTBC, ST, and STBC. (**b**). Load–displacement curves of the HTPC-1, HTPC-2, HTPC-3, and HHT. (**c**) Load–displacement curves of the OHT, OHTBR, and HHT. (**d**). Load–displacement curves of the HT, HHT, HST, and HSHT. (**e**). Load–displacement curves of the CT, CTBH, CHT, and CHTBH. (**f**). Load–displacement curves of the HBVT-257, HUT/HBVT-555, and HBVT-752.

**Figure 7 biomimetics-09-00060-f007:**
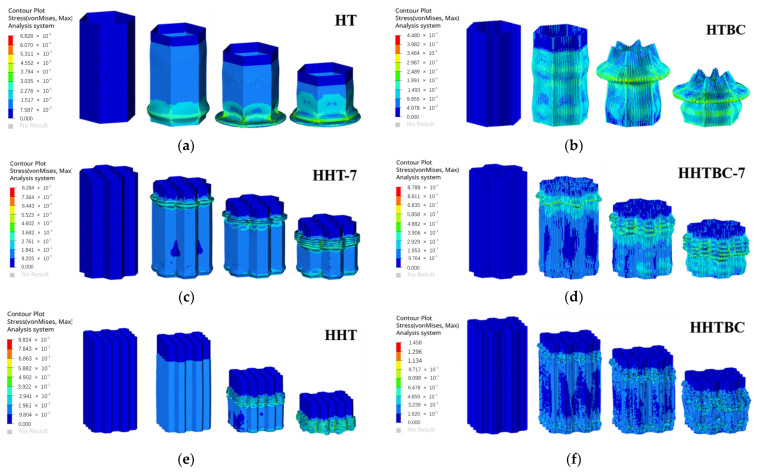
Axial compression deformation and stress cloud in honeycomb structures: (**a**). HT; (**b**). HTBC; (**c**). HHT-7; (**d**). HHTBC-7; (**e**). HHT; (**f**). HHTBC; (**g**). ST; (**h**). STBC; (**i**). HTPC-1; (**j**). HTPC-2; (**k**). HTPC-3; (**l**). OHT; (**m**). OHTBR; (**n**). HST; (**o**). HSHT; (**p**). CT; (**q**). CTBH; (**r**). CHT; (**s**). CHTBH; (**t**). HBVT-257; (**u**). HUT/HBVT-555; and (**v**) HBVT-752.

**Figure 8 biomimetics-09-00060-f008:**
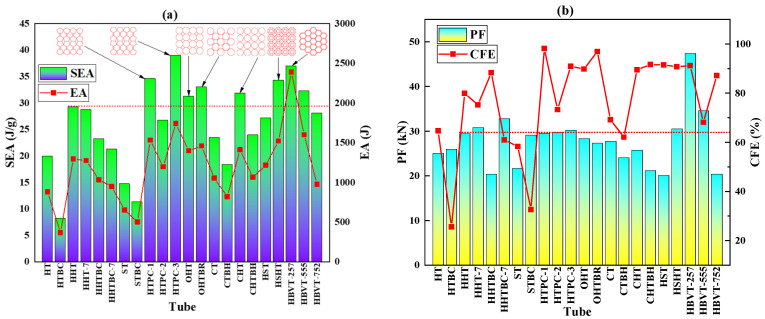
The energy absorption index of each structure under axial impact: (**a**). SEA and EA; and (**b**). PF and CFE.

**Figure 9 biomimetics-09-00060-f009:**
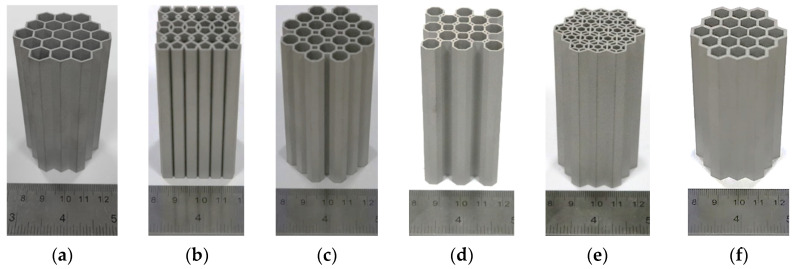
Three-dimensionally printed honeycomb structures: (**a**). HBVT-555; (**b**). HTPC-3; (**c**). OHT; (**d**). OHTBR; (**e**). HSHT; and (**f**). HBVT-257.

**Figure 10 biomimetics-09-00060-f010:**
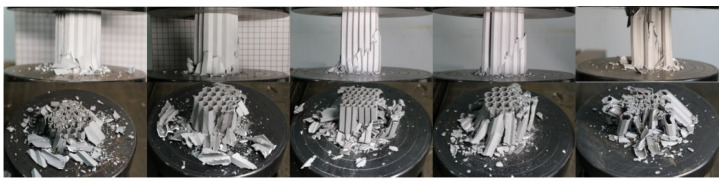
Part of the 3D-printed honeycomb structure deformation process.

**Table 1 biomimetics-09-00060-t001:** Dimensions and mass parameters of these honeycomb structures and energy absorption characteristics under axial impact load.

No.	Sample	Thickness/mm	Mass/g	EA/J	SEA/J/g	Fmax/kN	Fmean/kN	CFE/%
1	HT	1.52	44.26	891.25	20.14	25.06	17.825	71.13
2	HTBC	0.72	44.29	364.86	8.24	25.91	7.2972	28.16
3	HHT-7	0.81	44.31	1274.67	28.77	30.81	25.4934	82.74
4	HHTBC-7	0.61	44.40	1100.76	24.79	32.82	22.0152	67.08
5	HHT	0.51	44.28	1294.77	29.24	29.46	25.8954	87.90
6	HHTBC	0.34	44.38	1157.32	26.08	20.16	23.1464	114.81
7	ST	1.28	44.24	693.858	15.68	21.63	13.87716	64.16
8	STBC	0.82	44.25	522.401	11.81	29.09	10.44802	35.92
9	HTPC-1	0.45	44.27	1593.38	35.99	29.52	31.8676	107.95
10	HTPC-2	0.43	44.70	1196.78	26.77	29.70	23.9356	80.59
11	HTPC-3	0.44	44.62	1826.39	40.93	30.13	36.5278	121.23
12	OHT	0.39	44.66	1397.5	31.29	28.30	27.95	98.76
13	OHTBR	0.45	44.15	1459.85	33.07	27.38	29.197	106.64
14	HST	0.42	44.71	1215.82	27.19	20.03	24.3164	121.40
15	HSHT	1.81	44.39	1525.74	34.28	30.56	30.5148	99.85
16	CT	1.65	44.78	1053.49	23.53	27.69	21.0698	76.09
17	CTBH	1.75	44.55	820.196	18.41	24.02	16.40392	68.29
18	CHT	0.39	44.30	1411.81	31.87	25.37	28.2362	111.30
19	CHTBH	0.43	44.41	1065.09	23.98	21.14	21.3018	100.77
20	HBVT-257	0.25/0.5/0.75	53.73	2376.98	37.02	47.40	47.5396	100.29
21	HBVT-555	0.51	44.28	1294.77	29.24	34.55	25.8954	74.95
22	HBVT-752	0.75/0.5/0.25	34.83	978.51	28.23	20.42	19.5702	95.84

**Table 2 biomimetics-09-00060-t002:** The axial compression energy absorption characteristics of each sample.

No	Sample	Mass/g	EA/J	SEA/J/g
1	HBVT-555	64	1609.60	25.15
2	HBVT-257	75	2124.00	28.32
3	HBVT-752	105	2573.55	24.51
4	HTPC-3	85	2625.65	30.89
5	OHT	77	1973.51	25.63
6	OHTBR	108	1995.92	27.74
7	HSHT	125	3478.75	27.83

## Data Availability

Other researchers can access the data supporting the conclusions of the study. (1) The nature of the data is the source data of the image in the paper; (2) The data can be accessed on the submitting system or emailed to iansongjiafeng@163.com; (3) There are no restrictions on the data access.

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
