# Peer review of "Biomimetic Study of a Honeycomb Energy Absorption Structure Based on Straw Micro-Porous Structure"

_biomimetics, 2024, doi:10.3390/biomimetics9010060_

Round 1

Reviewer 1 Report

Comments and Suggestions for Authors

The manuscript reports an experimental-numerical study on honeycomb energy absorption structure inspired by straw micro-porous structure. Based on the reported results, the specific energy absorption of bio-inspired design is about 20% higher than that of the equivalent traditional structure. The topic is interesting and the subject perfectly falls into the scope of the journal. However, the following points should be carefully addressed before any consideration for publication.

1.     Please update and enrich the abstract to reflect and highlight the major findings and main contribution more clearly.

2.     "Bionic design" is listed as a keyword, however, it is not used in the abstract. Please update accordingly.

3.     For the multi-window figures use the same format in the entire manuscript. The authors used numeric, alphabetical and direct captioning in different figures, which is not acceptable. Please use uniform method for all the figures based on the journal's requirements.

4.     It is not clear what the procedure of the impact tests is, please provide more detail for the results reported in Table 1.

5.     It's unclear why two different materials were used in the experimental and numerical studies, which hinders the meaningful comparison of results. It would greatly enhance the manuscript if the same material used in 3D printing for the experiments was also used in the numerical model. The difference in materials used in the experiments and simulations results in varying failure modes and energy dissipation capacities, making direct comparisons challenging. Addressing this discrepancy would improve the validity of the study and the relevance of the results.

6.     With an average performance enhancement of approximately 20%, it's worth considering whether the advanced techniques required for these bio-inspired designs compared to traditional sandwich structures justify their economic viability. In other words, it's essential to elucidate the specific applications where the suggested structures outperform traditional models and address problems that cannot be efficiently solved through conventional methods. This information will help in understanding the practical utility and cost-effectiveness of the proposed methodology.

7.     The literature review in the manuscript appears to be outdated, as it lacks references to papers published in the last two years. Considering that there are numerous papers on bio-inspired sandwich structures published annually, it is crucial to update the literature review to reflect the current state of the art in this field. This will ensure that the manuscript is up-to-date and includes relevant references to the latest research.

8.     In the experimental program not enough details are provided for the testing procedure, loading and measuring methodology. In this regard an update is necessary.

9.     Referring to the FE model, the authors should provide all necessary details to ensure the reproducibility of FE results. For example, what is the mesh size and element type? What is the loading procedure? What is the applied velocity? Did the author consider any type of imperfection in the model?

10.  Please provide a reference for the statement in page 7 that says aluminum is strain-rate insensitive material.

11.  Is MAT_24 in LS-DYNA able to predict the crushing, tearing and separation? Ignoring that two different material is used in experimental and numerical study, can the adopted material model predict the behavior observed in the experiment?

12.  Please show time-history of the impact load in a diagram.

13.  In Figure 7, it's important to clarify which stress contour is displayed in the images. Without a legend, the colors are not meaningful. Additionally, given that it is a dynamic analysis, it should be indicated which specific time point these results correspond to. To enhance the clarity and usefulness of the figure, you may consider adding more details, such as a legend, or alternatively, you can choose to remove the stress contours and present only the deformed shape. This will make the figure more informative and comprehensible for readers.

Author Response

The manuscript reports an experimental-numerical study on honeycomb energy absorption structure inspired by straw micro-porous structure. Based on the reported results, the specific energy absorption of bio-inspired design is about 20% higher than that of the equivalent traditional structure. The topic is interesting and the subject perfectly falls into the scope of the journal. However, the following points should be carefully addressed before any consideration for publication.

  1. Please update and enrich the abstract to reflect and highlight the major findings and main contribution more clearly.

Answer: Thank you for your valuable feedback. We have revised and enriched the abstract to better reflect the major findings and main contributions of our study.

In the abstract, we have added the following content: "According to the characteristics of two types of straw microcell and chamber structures, a cellular energy absorption structure for bionic optimization design is proposed. A total of 22 types of honeycomb structures are classified into 6 categories, including cell wall corrugated-type bionic design, modular cell design, reinforcement plate structure, self-similar structure, porous structure of cell wall, and gradient structure of variable wall thickness. Among them, HTPC-3 (combined honeycomb structure), HSHT (self-similar honeycomb structure), and HBCT-257 (radial gradient variable wall thickness honeycomb structure) exhibit the best performance, with their energy absorption being 41.06%, 17.84%, and 83.59% higher than that of HHT (traditional hexagonal honeycomb decoupling unit), respectively. Moreover, the specific energy absorption is increased by 39.98%, 17.24%, and 26.61%, respectively, compared to HHT. Through verification test analysis, the performance of combined honeycomb structure is the best, and its specific energy absorption is 22.82% higher than that of the traditional hexagonal structure."

  1. "Bionic design" is listed as a keyword, however, it is not used in the abstract. Please update accordingly.

Answer: We appreciate the feedback. We have revised the abstract to include the term "bionic design", as shown in “Abstract”.

  1. For the multi-window figures use the same format in the entire manuscript. The authors used numeric, alphabetical and direct captioning in different figures, which is not acceptable. Please use uniform method for all the figures based on the journal's requirements.

Answer: We have ensured uniformity in the format of multi-window figures throughout the manuscript, following the journal's requirements. Refer to Figure 1, Figure 2, Figure 3, Figure 6, Figure 7 and Figure 8.

  1. It is not clear what the procedure of the impact tests is, please provide more detail for the results reported in Table 1.

Answer: In response to the concern raised about the lack of clarity regarding the procedure of the impact tests, we have provided additional details for the results reported in Table 1. Specifically, we have included a more comprehensive description of the impact test procedure, highlighting key parameters and methodologies.

With reference to “Table 1”, the detailed information is as follows: “The mass of the hammer used in the experiment was 100 kg for each specimen, and the impact velocity was 6.85 m/s”.

  1. It's unclear why two different materials were used in the experimental and numerical studies, which hinders the meaningful comparison of results. It would greatly enhance the manuscript if the same material used in 3D printing for the experiments was also used in the numerical model. The difference in materials used in the experiments and simulations results in varying failure modes and energy dissipation capacities, making direct comparisons challenging. Addressing this discrepancy would improve the validity of the study and the relevance of the results.

Answer: The validation experiments in this paper use stainless steel, considering its optimal mechanical stability in current 3D metal printing technology. Additionally, the material model set in the validation experiments aligns with that used in the corresponding simulation. This consistency indicates the effectiveness of the simulation approach in this study.

  1. With an average performance enhancement of approximately 20%, it's worth considering whether the advanced techniques required for these bio-inspired designs compared to traditional sandwich structures justify their economic viability. In other words, it's essential to elucidate the specific applications where the suggested structures outperform traditional models and address problems that cannot be efficiently solved through conventional methods. This information will help in understanding the practical utility and cost-effectiveness of the proposed methodology.

Answer: First, the energy absorption performance of the mentioned energy-absorbing structure in this paper is improved compared to traditional honeycomb structures.

Second, the honeycomb structure proposed in this paper can be mass-produced and manufactured through an extrusion-based manufacturing method, providing certain economic viability and diverse industrial applications.

  1. The literature review in the manuscript appears to be outdated, as it lacks references to papers published in the last two years. Considering that there are numerous papers on bio-inspired sandwich structures published annually, it is crucial to update the literature review to reflect the current state of the art in this field. This will ensure that the manuscript is up-to-date and includes relevant references to the latest research.

Answer: Following expert recommendations, we have added references related to the latest honeycomb energy absorption structures from 2023 and 2024. The specific references are as follows:

JJIAPENG S, YULONG HE, XIUJUAN Z, XIN L, MINGHUI L, YANFENG C.Energy absorption and topology optimization of self-similar inspired multi-cell square tubes[J].Thin-Walled Structures,2024,Vol.196: 111491.

JUNDONG Z, RUIYAO L, XIANG L, QING C, ZHIYING W, YUNTING G, ZHIXIN L, QI Z, ZEZHOU X, GUOFENG Y, LUQUAN R. Characteristic analysis of bionic-induced structures with negative stiffness inspired by the growth and deformation differences of branches [J].Thin-Walled Structures, 2024, 196: 111437.

  1. In the experimental program not enough details are provided for the testing procedure, loading and measuring methodology. In this regard an update is necessary.

Answer: we have revised the experimental section to include more comprehensive information on the testing procedure, loading conditions, and the methodology used for measurements.

With reference to “4.1. Processing and manufacturing”, the detailed information is as follows: “To better observe the deformation modes of honeycomb structure with different biomimetic cells, quasi-static axial compression tests were conducted using the ETM-300 (with a maximum load capacity of 300 kN) at the Key Laboratory of Engineering Bionics, Ministry of Education, Jilin University. All designed samples were tested at standard room temperature (25°C). The test loading speed was set at 3 mm/min, and the final loading displacement was 2/3 of the sample height. Load and displacement data were obtained from the data acquisition system. During the experiment, deformation modes during compression were recorded using a camera.”.

  1. Referring to the FE model, the authors should provide all necessary details to ensure the reproducibility of FE results. For example, what is the mesh size and element type? What is the loading procedure? What is the applied velocity? Did the author consider any type of imperfection in the model?

Answer: We appreciate the insightful feedback from the reviewers, and we understand the importance of providing sufficient details for the Finite Element (FE) model to ensure the reproducibility of our results.

In response to these comments, we have included comprehensive information on the FE model, addressing the following key aspects: In the Optistruct module of Hypermesh software, tetrahedral meshing was performed on the honeycomb energy absorption structure. After adjusting element sizes, splitting, and other operations, a mesh with approximately 120,000 nodes and 100,000 non-distorted C3D4 elements was obtained.

  1. Please provide a reference for the statement in page 7 that says aluminum is strain-rate insensitive material.

Answer: Yeonju N, Min-Su L, Umer M C, Tea-Sung J. Effect of strain rate on the deformation of 6061-T6 aluminum alloy at cryogenic temperature [J]. Materials Characterization, 2023, 206: 113403.

Manes A, Peroni L, Scapin M, Giglio M. Analysis of strain rate behavior of an Al 6061 T6 alloy [J]. Procedia Engineering, 2011, 10: 3477-3482.

  1. Is MAT_24 in LS-DYNA able to predict the crushing, tearing and separation? Ignoring that two different material is used in experimental and numerical study, can the adopted material model predict the behavior observed in the experiment?

Answer: The material fracture in the experiment mainly stems from errors generated during manufacturing. As MAT24 does not incorporate a fracture constitutive model for the material, our focus in presenting the effectiveness of the simulation model is more on the lateral comparison between experimental samples and the comparison of data within the simulation environment.

  1. Please show time-history of the impact load in a diagram.

Answer: Since the validation experiment is quasi-static, the curves of the time history and displacement history are essentially consistent.

  1. In Figure 7, it's important to clarify which stress contour is displayed in the images. Without a legend, the colors are not meaningful. Additionally, given that it is a dynamic analysis, it should be indicated which specific time point these results correspond to. To enhance the clarity and usefulness of the figure, you may consider adding more details, such as a legend, or alternatively, you can choose to remove the stress contours and present only the deformed shape. This will make the figure more informative and comprehensible for readers.

Answer: Thank you for the valuable feedback. In Figure 7, we have incorporated a detailed legend to clearly indicate the stress contour displayed in the images.

Reviewer 2 Report

Comments and Suggestions for Authors

The research work by Xu et al aims to investigate the potential in terms of energy absorption of biomimetic prototypes of natural origin. In particular, the authors focused attention on light structures derived from two kinds of straw: sorghum and reed. The research carried out through both theoretical and experimental analyzes is certainly relevant given the numerous industrial applications of energy absorption devices. However, the text requires mainly linguistic revisions to improve its understanding by a wide range of readers and the rapid transfer of the acquired know-how to professionals.

Among the points to be reviewed, the following are mentioned:

Page 3, Paragraph 2: This section of the test reports sentences in a list-like manner (lines 5-9). It is suggested to report the contents in a conversational manner so as not to make the reading of the contents burdensome.

Page 4, Paragraph 2.2, line 2: the text reports a sentence that begins with "Similar to bamboo nodule structure, ..." but this concept is already included in the previous sentence. Please check to avoid unnecessary repetitions.

Page 5, Paragraph 2.3, line 4: the authors write "In this paper, it is simplified into ...", the subject is missing? Otherwise, review the sentence structure to improve understanding. Furthermore, the same sentence that continues to the end of the paragraph is too long. It is advisable to review the construction of this sentence to reduce its length and facilitate reading.

Page 6, Paragraph 2.6, lines 5-6: the text reports the sentence starting with "They are deviated into two types, ..." but it is not clear who the subject of this sentence is. Please check and revise the sentence.

Page 10, line 26: correct the text "...expecially HBVT-257..." to read "... expecially for HBVT-257...".

Page 13, Paragraph 4.2, lines 7-8: what is meant by "material problems"? Revise the sentence to eliminate repetitions of the word "materials".

Page 14, Paragraph 5: Conclusions begin with a very long sentence (five lines). Similarly to what has already been commented, please review the construction of this sentence to facilitate the learning of the concepts for all readers.

Comments on the Quality of English Language

The English language must be carefully revised,

Author Response

  1. The research work by Xu et al aims to investigate the potential in terms of energy absorption of biomimetic prototypes of natural origin. In particular, the authors focused attention on light structures derived from two kinds of straw: sorghum and reed. The research carried out through both theoretical and experimental analyzes is certainly relevant given the numerous industrial applications of energy absorption devices. However, the text requires mainly linguistic revisions to improve its understanding by a wide range of readers and the rapid transfer of the acquired know-how to professionals.

Among the points to be reviewed, the following are mentioned:

Page 3, Paragraph 2: This section of the test reports sentences in a list-like manner (lines 5-9). It is suggested to report the contents in a conversational manner so as not to make the reading of the contents burdensome.

Answer: We have revised the sentence, and the revised sentence is as follows: Zeiss scanning electron microscope (SEM, Model EVO-18, Germany) was employed. The main parameters of the SEM are as follows: the experimental magnification range is 13-50,000 times; the minimum resolution is 3.0 nm; the instrument utilizes a tungsten filament.

  1. Page 4, Paragraph 2.2, line 2: the text reports a sentence that begins with "Similar to bamboo nodule structure, ..." but this concept is already included in the previous sentence. Please check to avoid unnecessary repetitions.

Answer: Thank you, we have deleted the sentence of “Similar to bamboo nodule structure”.

  1. Page 5, Paragraph 2.3, line 4: the authors write "In this paper, it is simplified into ...", the subject is missing? Otherwise, review the sentence structure to improve understanding. Furthermore, the same sentence that continues to the end of the paragraph is too long. It is advisable to review the construction of this sentence to reduce its length and facilitate reading.

Answer: We've polished the language and streamlined the sentences.

  1. Page 6, Paragraph 2.6, lines 5-6: the text reports the sentence starting with "They are deviated into two types, ..." but it is not clear who the subject of this sentence is. Please check and revise the sentence.

Answer: We've polished the language.

  1. Page 10, line 26: correct the text "...expecially HBVT-257..." to read "... expecially for HBVT-257...".

Answer: We have modified it according to the reviewer's opinion.

  1. Page 13, Paragraph 4.2, lines 7-8: what is meant by "material problems"? Revise the sentence to eliminate repetitions of the word "materials".

Answer: Here, it is meant to convey that part of the difference between the experimental and simulated phenomena is due to objective factors such as material purity and processing density during the manufacturing process.

  1. Page 14, Paragraph 5: Conclusions begin with a very long sentence (five lines). Similarly to what has already been commented, please review the construction of this sentence to facilitate the learning of the concepts for all readers.

Answer: We've polished the language. As shown in following: In this paper, based on the characteristics of two types of straw microcell and chamber structures, a cellular energy absorption structure is proposed for bionic optimization design. A total of 22 honeycomb structures are classified into 6 categories, including bionic design with corrugated cell walls, modular cell design, reinforcement plate structure, self-similar structure, porous structure of cell walls, and gradient structure with variable wall thickness.

Round 2

Reviewer 1 Report

Comments and Suggestions for Authors

The author has effectively addressed the primary concerns, and the manuscript has been sufficiently improved to warrant its acceptance.

It seems there is some formatting problems regarding used (u) and (v) in pages 8 and 9, please update accordingly.